# Evaluating the efficacy of the HITSystem 2.1 to improve PMTCT retention and maternal viral suppression in Kenya: Study protocol of a cluster-randomized trial

**Sharon Mokua**[1]*, **May Maloba**[2], **Catherine Wexler**[3], **Kathy Goggin**[4,5], **Vincent Staggs**[4], **Natabhona Mabachi**[3], **Nicodemus Maosa**[2], **Shadrack Babu**[2], **Emily Hurley**[4], **Sarah Finocchario-Kessler**[3]

**1** Kenya Medical Research Institute, Nairobi, Kenya, **2** Global Health Innovations–Kenya, Nairobi, Kenya, **3** Department of Family Medicine, University of Kansas Medical Center, Kansas City, Missouri, United States of America, **4** Children's Mercy Kansas City, Health Services and Outcomes Research, Kansas City, Missouri, United States of America, **5** University of Missouri-Kansas City, School of Medicine, Kansas City, Missouri, United States of America

* mokuasharon91@gmail.com, smokua@go.ke

**Data Availability Statement:** No datasets were generated or analysed during the current study. All

## Abstract

### Background

Gaps in the provision of guideline-adherent prevention of mother-to-child transmission of HIV (PMTCT) services and maternal retention in care contribute to nearly 8000 Kenyan infants becoming infected with HIV annually. Interventions that routinize evidence-based PMTCT service delivery and foster consistent patient engagement are essential to eliminating mother-to-child transmission of HIV. The HITSystem 2.1 is an eHealth intervention that aims to improve retention in PMTCT services and viral load monitoring, using electronic alerts to providers and SMS to patients. This study will evaluate the impact, implementation, and cost-effectiveness of HITSystem 2.1.

### Method

This cluster randomized trial will be conducted at 12 study hospital (6 intervention, 6 control). Pregnant women living with HIV who have initiated PMTCT care ≤36 weeks gestation are eligible. Women enrolled at control hospitals will receive standard-of-care PMTCT services. Women enrolled at intervention hospitals will receive standard-of-care PMTCT services plus enhanced HITSystem 2.1 tracking. Mixed logistic regression models will compare the arms on two primary outcomes: (1) completed guideline-adherence PMTCT services and (2) viral suppression at both delivery and 6 months postpartum. We will assess associations between provider and patient characteristics (disclosure status, partner status, depression, partner support), PMTCT knowledge, and motivation with retention outcomes. Using the RE-AIM model, we will also assess implementation factors to guide sustainable scale-up. Finally, a cost-effectiveness analysis will be conducted.

relevant data from this study will be made available upon study completion.

**Funding:** This study was funded by the National Institute of Mental Health.The study grant number is R01MH121245. The funders had and will not have a role in study design, data collection and analysis, decision to publish, or preparation of the manuscript.

**Competing interests:** The authors have declared that no competing interests exist.

**Abbreviations:** AIDS, Acquired immune deficiency syndrome; ANC, Antenatal care; ART, Antiretroviral therapy; CHW, Community health worker; cRCT, cluster randomized control trial; EID, Early infant diagnosis; HITSystem, HIV Infant Tracking System; HIV, Human immunodeficiency virus; IPV, Intimate partner violence; KES, Kenyan shillings; LTFU, Loss to follow up; MTCT, Mother to child transmission; PEPFAR, President's Emergency Plan for AIDS Relief; PMTCT, Prevention of mother to child transmission; SMS, short message services; SOC, Standard of care; VL, Viral load; WHO, World Health Organization.

## Discussion

This study will provide insights regarding the development and adaptation of eHealth strategies to meet the global goal of eliminating new HIV infections in children and optimizing maternal health through PMTCT services. If efficacious, implementation and cost-effectiveness data gathered in this study will guide scale-up across Kenyan health facilities.

## Trial registration

This study was registered at clinicaltrials.gov (NCT04571684) on October 1, 2020.

## Introduction

Prevention of mother-to-child transmission of HIV (PMTCT) programs provide services for pregnant women living with HIV through the antenatal, delivery, and postpartum phases and are critical in reducing mother-to-child transmission. All pregnant or breastfeeding women living with HIV in Kenya are eligible to receive life-long antiretroviral therapy (ART) [1].

By 2020, the proportion of pregnant Kenyan women living with HIV who were on ART had reached 94% [1]. However, a survey of records of 4 Kenyan hospitals from 2013 to 2016 indicated that only 38% of pregnant women living with HIV received ART regimens compliant with guidelines at the time [2]. Retention in care, particularly in the postpartum period, continues to pose a challenge. One study suggests 28% attrition through 6 months postpartum [3]. These Kenya data are consistent with broader literature highlighting suboptimal patient retention throughout the phases of PMTCT, including high rates of patient disengagement or loss to follow-up [4–10], poor maternal medication adherence [11–13], ART failure [14, 15], and missed opportunities for early infant diagnosis (EID) [16].

Maternal viral load (VL) levels provide a clear benchmark to guide clinical management and minimize risk of mother-to-child transmission [17]. Pregnant women with suppressed VL have very low risk of transmitting the virus to their infant. Clinical intervention to address unsuppressed maternal VL during pregnancy or breastfeeding can include enhanced adherence support, regimen switching, and enhanced infant prophylaxis [18–20]. These interventions are only feasible if eligible patients receive VL tests, results are promptly returned to health facilities, and detectable VL results lead to clinical action. Records reviewed at 4 Kenyan hospitals indicate insufficient VL testing among pregnant and postpartum women: only 72% received any VL test by 6 months postpartum and only 6% received guideline adherent baseline and repeat testing through 6 months postpartum. Furthermore, among patients who received a viral load test, few (36.0%) were notified of their results and only 28.9% had any documented evidence of clinical action taken [21]. Delays in timely clinical action after detectable VL results were cited in PEPFAR's Kenya Country Operational 2017 Plan as a key barrier in Kenya, and as a target for interventions [22].

Interventions that foster consistent patient engagement and routinize the delivery of best practices across all phases of PMTCT are essential to prevent perinatal HIV infections. HIT-System 2.0 is a web-based intervention that tracks pregnant women living with HIV and their infants to improve the completeness and efficiency of PMTCT and EID services. In a pilot study at two Kenyan hospital (1 control, 1 intervention), HITSystem 2.0 participants were more likely to receive complete PMTCT services compared to SOC (56.6% vs. 17.1% p < 0.001); however, viral load testing at both sites was suboptimal, prompting a HITSystem modification to more comprehensively prompt and track guideline-adherence viral load testing (HITSystem 2.1) [23]. The HITSystem 2.1 intervention aims to facilitate complete PMTCT retention and VL monitoring with prompt clinical action (adherence support, ART regimen

change) in the antenatal, delivery, and 6-month postpartum periods to increase viral suppression during windows critical for HIV prevention.

This study uses a cluster randomized controlled design at 12 Kenyan government hospitals with the overall goal of evaluating the HITSystem 2.1 intervention's impact on complete, guideline-adherent PMTCT care through 6 months postpartum. HITSystem 2.1 reflects the 2018 Kenyan PMTCT guidelines [24], including added intervention features that support routine viral load monitoring and interventions to optimize maternal ART adherence. We aim to rigorously evaluate the impact of HITSystem 2.1 with the long-term goal of optimizing the provision of guideline-adherent services and viral suppression through the antenatal, delivery, and early postpartum periods.

## Materials and methods

### Study overview

This is a 5-year cluster-randomized control trial designed to evaluate the HITSystem 2.1 in Kenya, while also assessing implementation to guide long term sustainability (NCT04571684). There are four specific objectives that this study aims to achieve: (1) evaluate the efficacy of HITSystem 2.1 in increasing the proportion of women who receive complete PMTCT services through 6 months postpartum, per Kenyan national guidelines, (2) evaluate the efficacy of HITSystem 2.1 in increasing viral suppression (<1,000 ml/copies) through 6 months postpartum, (3) evaluate the reach, effectiveness, adoption, implementation fidelity, and maintenance (RE-AIM model) of HITSystem 2.1 using system data and surveys with providers and patients, and (4) evaluate the costs and cost-effectiveness of HITSystem 2.1 for increasing complete PMTCT retention, viral suppression, and modeled estimates for pediatric HIV infections averted, see **Fig 1** for the schedule of enrollment, interventions and assessments [25]. All study procedures were approved by Institution Review Boards at the University of Kansas Medical Center (STUDY00144753) on December 5[th] 2019 and Kenya Medical Research Institute (SERU3983) on 27[th] April 2020. At the time of PMTCT enrollment, all eligible women will be informed about the study by research staff, and provide written informed consent for participation.

### HITSystem 2.1 intervention

The HITSystem is accessed online through a computer or tablet, using mobile broadband modems that respond to cellular signal. There are two active modules of the HITSystem: the EID module of the HITSystem (HITSystem 1.0) combines SMS outreach to mothers of HIV-exposed infants and algorithm-based dashboard alerts to prompt provider (MCH/HIV and laboratory) action to improve retention of mother-infant pairs in early infant diagnosis of HIV (EID) services and ART initiation among HIV-infected infants [26]. The PMTCT module of the HITSystem (HITSystem 2.0) expands the scope of the intervention to involve pregnant women living with HIV, and primary intervention components include: (1) electronic prompts to notify PMTCT providers and program managers when action is required to support timely services (ART initiation, syphilis testing, linkage to EID) or when patients have missed appointments and (2) automated text messages to participants' mobile phones with the aim of motivating adherence to medication, reminding women of ANC appointments and medication refills, and prompting preparation for a hospital delivery [23]. A recent modification to the PMTCT module (HITSystem 2.1) has increased VL test tracking and follow up and has expanded the duration of maternal follow up from 6 weeks to 6 months. Automatic linkage from the PMTCT module to the EID module allows for simplified tracking of mother-infant pairs from pregnancy through EID completion at 18 months. Table 1 outlines the progression of the HITSystem from 1.0 to 2.1.

| Timepoint | -t1 Pre-Enrollment | Post enrollment | | | Close out |
| | | t1 Baseline | t2 Antenatal period | t3 Delivery (0-4w postpartum) | t4 Postpartum period (4w-7m PP) |
|---|---|---|---|---|---|
| **Enrollment** | | | | | |
| Allocation[a] | X | | | | |
| Eligibility Screen | X | | | | |
| Informed consent | X | | | | |
| **Interventions** | | | | | |
| Baseline survey | | X | | | |
| SOC viral load testing | | X | X | X | X |
| SOC PMTCT service provision | | X | X | X | X |
| HITSystem tracking, follow up[b] | | X | X | X | X |
| Delivery viral load test | | | | X | |
| Delivery survey | | | | X | |
| Postpartum survey | | | | | X |
| Postpartum VL test | | | | | X |
| Participant tracking for VL tests | | | | X | X |
| **Assessments** | | | | | |
| Obj 1: Complete PMTCT services | | X | X | X | X |
| Obj 1: Associated factors | | X | | X | X |
| Obj 2: VL at delivery | | | | X | |
| Obj 2: Postpartum VL | | | | | X |
| Obj 2: VL efficiency and utility | | X | X | X | X |

[a]allocation occurs at the facility level so will be known prior to eligibility screen and/or informed consent,
[b]intervention sites only
t1 – baseline
t2 – antenatal period
t3 – delivery (<4 wees postpartum)
t4 – postnatal period (4 weeks through 6m postpartum)
SOC= standard of care
VL= viral load

**Fig 1. SPIRIT schedule of enrollment, interventions, and assessments.**

**Table 1. Key difference between iterations of the HITSystem.**

| | HITSystem 1.0 | HITSystem 2.0 | HITSystem 2.1 |
|---|---|---|---|
| Services | EID | PMTCT | |
| Population | HIV-exposed infants | Pregnant women/mothers living with HIV | |
| Time period | 6 weeks– 18 months | ANC– 6wk postpartum | ANC– 6 month postpartum |
| Targeted outcomes | • Efficiency of sample processing<br>• ART initiation for HIV+ infants<br>• Retention through 18 months | • Appointment attendance<br>• ART adherence<br>• Hospital delivery<br>• Infant linkage to EID by 6wk postnatal | HITSystem 2.0, plus:<br>• VL testing through 6m postpartum<br>• Efficiency of VL testing<br>• Clinical action based on VL result |

EID = early infant diagnosis; PMTCT = prevention of mother-to-child transmission; VL = viral load; ART = antiretroviral therapy; ANC = antenatal care

## Site selection

The study will be conducted within twelve government health facilities in Siaya (n = 8), Kilifi (n = 2), and Mombasa (n = 2) counties of Kenya. These counties were selected due to their elevated perinatal HIV transmission rates and lower density of interventions targeting perinatal HIV transmission that could confound the results of this evaluation. To be eligible for selection and matching in the study, health facilities were required to provide PMTCT, EID, and ART services and not be involved in any ongoing PMTCT or EID interventions or research studies.

## Randomization

Randomization was conducted at the facility level. Study sites were matched on geographic region, resource level, and patient volume to produce six matched pairs. The study statistician randomly allocated one site in each pair to the intervention, using a random number generator program. The study statistician and all research staff were blinded to the randomization process. However, neither study staff nor participants will be blinded after the randomization process.

## Study staffing and training

A research assistant (RA) will be stationed at each study site and trained on tasks specific to their site designation (i.e., intervention vs control site) and will be overseen by a Kenyan study coordinator. Two study coordinators will oversee sites, based on geographic location and will be responsible for routine site visits. Table 2 outlines key responsibilities of RA and study coordinators. During the first weeks of implementation, the study coordinators will work closely with the RA and site-level clinical staff to ensure thorough understanding and accurate independent use of the HITSystem 2.1. Key hospital personnel (PMTCT and maternity nurses and mentor mothers) will be trained by employing hands-on data entry scenarios tailored to the specific capacity in which they will utilize the HITSystem 2.0.

## Aims 1 and 2 methods

Aims 1 and 2 will use the same patient population and similar methods; thus, they will be described together, see Fig 2.

**Participant eligibility and consent.**   All pregnant women living with HIV presenting for PMTCT services at study hospitals by 36 weeks gestation will be eligible for enrollment. Since Kenyan law allows all pregnant females the same capacity to consent as an adult, young women under the age of 18 will be able to provide consent for themselves and will be eligible for inclusion. Women will be excluded from study participation if they have any condition

**Table 2. Study staffing and activities at intervention and control sites.**

|  | Intervention site | Control site |
|---|---|---|
| Study Coordinator (n = 2, overseeing 12 sites) | • Provider surveys<br>• Monitor study data<br>• Facilitate home-based testing<br>• Routine visits |  |
|  | • HITSystem troubleshooting/ retraining | • Electronic entry of manually recorded standard-of-care clinical data |
| Research Assistant (n = 12, 1 per site) | • Participant screening<br>• Informed Consent<br>• Surveys: enrollment, delivery, postpartum<br>• Facilitate delivery VL tests |  |
|  | • HITSystem data entry and alert monitoring | • Support of SOC services without intervention |

**Fig 2. Patient flow through study services.** [Normal text represents standard PMTCT services, per Kenyan guidelines. Italicized text represents study-specific services.].

(including drug abuse, alcohol abuse, or psychiatric disorder) that study or hospital staff feel precludes their ability to provide informed consent. Capacity will be determined at the time of consent. Women who transfer care from one study site to another during their PMTCT services will be ineligible for enrollment at their new facility.

**Procedures.** *Baseline surveys* will be conducted at study enrollment at both intervention and control sites. The baseline survey will collect demographic and clinical data, as well as patient-level data regarding PMTCT information, motivation, partner status and support, HIV disclosure, depression (modified Edinburgh postnatal scale [27]), and risk of intimate partner violence (Appendix 1 in S1 Appendix). Repeated surveys after delivery and 6m postpartum will include these same patient-level measures to assess potential correlates of outcomes (Appendices 2, 3 in S1 Appendix). Surveys are estimated to take about 15 minutes and participants will be compensated 200 KES for completion of each survey.

**Clinical data collection.** *At sites implementing HITSystem 2.1*, all patient-specific data including: demographics; women's phone number and patient tracing information; women's appointment attendance and ART medication refills/pill counts; dates of maternal VL sample collection, test processing, results return, and notification; maternal VL results; infant's date of birth, birth weight, and gestational age at delivery will be entered directly into HITSystem 2.1, as they become available. At enrollment, intervention site participants will indicate their SMS preferences by opting in or out of SMS for appointment reminders (sent 2 days before scheduled appointment), delivery support (sent 4 weeks and 2 weeks prior to estimated delivery date and encourages planning for a hospital-based delivery), and ART adherence (option to select never, daily, weekly, monthly).

*At control sites*, all clinical and PMTCT service-related data will be recorded in the existing paper-based registries by health care providers as part of routine services. To prevent stimulating unintended intervention at control sites because of more comprehensive review of clinical data for research, site coordinators–rather than site-based research assistants–will collect control site data. To ensure comparable data collection tools and quality at intervention and control sites, control site data will be entered into an electronic system that mirrors the HITSystem's data collection fields but has all HITSystem functionality (provider alerts and SMS features) disabled.

*Viral load tracking*: To understand the efficacy of HITSystem 2.1 to increase viral suppression we will systematically measure VL of all women (intervention and control) at the time of delivery (defined as within 4 weeks of delivery) and by 7 months postpartum to approximate maternal VL during the antenatal and early postnatal periods. VL testing at delivery goes beyond the guidelines and is done solely for research purposes. To ensure that this testing does not interfere with our ability to examine routine VL testing postpartum and avoid confusion regarding VL guidelines, the delivery VL test will be coordinated by study staff, processed by the CDC KEMRI laboratory (not the clinical laboratory) and results will be maintained outside of clinic files by the US-based study manager. For the postpartum VL test, results of the first routine postpartum VL test will be collected from the medical record. Per guidelines, the first postpartum VL test should occur within 6 months, so even women tested very late in pregnancy will be due for retesting by 6 months postpartum. Participants without a repeat VL test by 5 months postpartum will be tracked by study staff through phone or in-person contact to schedule a VL test by 7 months postpartum.

*Participant tracking for VL among participants 'lost to follow-up' (LTFU)*. Measurement of viral suppression rates in either study arm can be skewed if detectable VL is more/less frequent in LTFU patients. Therefore, we will track patients who receive no delivery VL test or postpartum VL test within the indicated time windows from a "watchlist" that was developed for each site to indicate patients in need of a time-sensitive VL test. Women without a routine VL by

the indicated time will be targeted for active tracking including phone calls and home visit. Study staff conducting home visits (mentor mother and community health worker) will be thoroughly trained on protection of confidentiality, especially when participants are not home or have relocated. Visits will be introduced as well-baby visits to protect confidentiality, and a dried blood spot sample (DBS) for maternal VL will only be collected once privacy is confirmed and consent provided. Visits will be rescheduled if privacy is perceived to be compromised at any time. (Appendix 4 in S1 Appendix). Participants who re-engage in care after active tracing for viral load collection will not be eligible to be considered "completely" retained in care.

*Potential correlates of HITSystem efficacy* will be assessed through patient characteristics gathered in patient surveys at baseline, delivery, and at the postpartum viral load test. Facility assessment forms (Appendix 5 in S1 Appendix) will be completed at the beginning, mid-point and end of study implementation to assess facility factors such as average resource level, annual PMTCT volume, and PMTCT provider-patient ratios.

*Provider surveys* will be conducted at baseline and again at the end of the study period. Providers in both arms will be asked to complete a short survey (Appendix 6 in S1 Appendix) assessing provider role/ department, level of experience, knowledge and motivation regarding current PMTCT guidelines, and barriers and facilitators to provision of complete PMTCT services. Providers will be eligible to complete the survey if they (1) work primarily in ANC, PMTCT, CCC, laboratory, or MCH departments, (2) will/have been involved in the provision of PMTCT care during the course of the study, and (3) have interacted with the HITSystem 2.1 during the course of the study (intervention sites, only).

*Completion of study follow up*. Study follow-up is complete when participants receive their postpartum viral load test and complete the postpartum survey OR when they reach 7 months postpartum. Participants at both intervention and control sites will continue to receive PMTCT and EID services per their hospital's standard of care. Participants from intervention sites can choose to continue receiving enhanced follow up for their infant through the EID module of the HITSystem. Patient clinical data after these completion points will not be documented as part of the study.

## Aim 1 outcomes and analyses

**Aim 1 outcomes.**   The primary outcome for Aim 1 is receiving "complete PMTCT services," an aggregate measure incorporating the documented receipt of all designated services across the antenatal (ART initiation, on-time appointment attendance, VL testing per guidelines, clinical action upon detectable VL), delivery (hospital-based delivery), and postpartum (EID linkage by 7 weeks, VL testing per guidelines, clinical action upon detectable VL) phases of PMTCT services. Participants who receive all indicated services per guidelines will be coded as 1 or 'yes'. Participants missing 1 or more services will be coded as 0 or incomplete PMTCT services.

Secondary outcomes associated with Aim 1 will include gestational week at PMTCT enrollment, duration of PMTCT retention (weeks), and ART adherence (proportion >95%, proportion with missing/late prescription refill). Proportions will be calculated to describe each component of "complete PMTCT services" separately (appointment attendance, VL monitoring, hospital delivery, EID linkage). Patient and facility-level factors will be assessed as correlates of receiving complete PMTCT. Provider characteristics will be aggregated and treated as facility-level variables. We will measure provider and patient information and motivation both pre- and post-intervention to examine changes in these variables throughout the study period. We also will measure the proportion and timing/age of any HIV transmissions to infants, and maternal and/or infant deaths.

*Statistical analyses for Aim 1 (complete PMTCT services)*. Data from women who have a documented reason for incomplete care at the study hospital (e.g., transfer facility, pregnancy loss, maternal death) will not be included in analyses. To demonstrate the efficacy of HITSystem 2.1 versus standard of care, we will test for differences between arms by modeling complete PMTCT as a function of study arm and selected covariates (e.g., age, partner support, depression) using logistic mixed models with a random site intercept (to adjust for clustering at the facility level). We will explore patient and facility-level characteristics as correlates of HITSystem effectiveness. We will compute average levels of provider knowledge and motivation for each site at baseline and end of study. These knowledge and motivation scores, as well as patient knowledge and motivation levels, will be compared pre-post intervention and between study arms to assess change within and between sites. In addition, provider and patient knowledge and motivation will be examined as predictors of PMTCT completion using logistic regression.

## Aim 2 outcomes and analyses

**Aim 2 outcomes.** The primary outcome for Aim 2 is suppressed VL at both delivery and within 6 months postpartum. Secondary outcomes associated with Aim 2 will include VL test coverage (VL test in antenatal and postpartum periods, per guidelines), VL test efficiency (turn-around time from sample to result and turn-around time from result to patient notification), VL test utility (detectable VL results with clinical action per guidelines, such as: intensified adherence counseling and/or ARV regimen change), and repeated VL testing among those with detectible VL. We will document outcomes for pregnant and breastfeeding women who are LTFU to better enumerate outcomes for this population.

**Statistical analyses for Aim 2.** We will test for differences on binary outcomes between arms using logistic mixed models with adjustment for clustering and select covariates. After transformation (e.g., log) of turn-around time values, if needed, differences in time will be assessed using appropriate generalized or linear mixed models. Post hoc analyses will assess person-level characteristics as moderators of the HITSystem 2.1 effect to assess whether the HITSystem 2.1 is more effective for specific subgroups. Any differences will guide future adaptations to improve efficacy among specific populations.

## Power and sample size

Sample size calculations were carried out to ensure sufficient power to detect a difference in odds of complete PMTCT services between the intervention (p1) and control (p2). We estimated current levels of retention in complete, guideline-adherent PMTCT services based on findings from our pilot work [23]. In the HITSystem R34 pilot study, guideline-adherent PMTCT services were completed at a much higher rate at the intervention facility (11% SOC v 43% HITSystem, p = 0.002, 32 percentage point increase). When complete, guideline-adherent PMTCT is defined to include VL monitoring, completion rate estimates drop substantially. Findings from our retrospective review at 4 Kenyan health facilities indicate only 50/424 (11.8%) of women enrolled in PMTCT services received both a guideline- adherent VL test in pregnancy AND a guideline adherent retest within 6 months postpartum. We conservatively anticipate HITSystem 2.1 supported VL testing will exceed that of standard of care by at least 20 percentage points (e.g., 12% v 32%). We assumed that the small subset of participants with guideline-adherent VL testing will be included among the larger population of those receiving complete PMTCT retention. To account for increased familiarity with guidelines over time and continually improving SOC, we assessed statistical power for contingencies with SOC guideline adherence as high as 18% using R package clusterPower [28]. Assuming ICC = .05,

the planned sample size of six 75-person clusters per arm provides 85% power to detect differ-
ences even smaller than the expected 20-percentage point difference across a range of possible
standard of care completion rates. Using HITSystem 2.0 pilot data, we predict 28% attrition;
thus, we need to enroll a minimum of n = 96 women per cluster (75 + 21). To account for pos-
sible temporal differences in enrollment, we will enroll at each pair of matched sites until
n = 96 is reached at both sites of the matched pair OR until a full year of enrollment data has
occurred, whichever is later.

## Aim 3 outcomes and analyses

We will evaluate the implementation of HITSystem 2.1 components to guide programmatic
implementation and increase sustainable use in government health facilities. The RE-AIM
framework is well suited to evaluate the implementation of system-level interventions, and
outlines the public health impact as the intersection between: Reach, Effectiveness, Adoption,
Implementation, and Maintenance. Table 3 outlines how each of these components of the
RE-AIM model will be evaluated.

**Statistical analyses for Aim 3.** Clinical and survey data will be summarized using descrip-
tive statistics and analyzed per Aim 1 and 2's analysis strategy. Qualitative content of field
notes (documentation of contextual events over the course of the study [i.e., strikes, reagent
stock outs, laboratory machine breakdowns], email exchanges with other relevant summary
documentation regarding implementation, meeting minutes of team calls) will be reviewed
and coded in summary themes relevant to the implementation process and experience, per the
RE-AIM model.

## Aim 4 specific methods

We will establish a unit associated with each type of resource used, estimate the dollar value of
each resource, calculate the number of units used, and multiply the units of resources used by
the dollar cost for each. Costs (licensing, computing/telecommunication, and personnel/train-
ing) will then be summed to derive the low, medium, and high estimates of the total cost of the
intervention. Research (research personnel, remuneration) costs will be calculated separately.
Cost data will be collected at the end of study years 2 & 4. We will then remove the costs saved
with HITSystem 2.1 from the gross program cost. We will conduct sensitivity analyses with a
range of estimates for HIV prevalence, rates of viral suppression, mother-to-child transmission

**Table 3. HITSystem 2.1 implementation evaluation guided by the RE-AIM model.**

|  | Definition | Measures | Data Sources |
|---|---|---|---|
| Reach | Number, proportion, representativeness of individuals who participate | • HITS 2.1 enrollment vs refusal<br>• Patient characteristics | • Enrollment log vs hospital registers<br>• Patient surveys |
| Effectiveness | The impact of an intervention on important outcomes | Aim 1 and Aim 2 primary outcomes | • HITS 2.1<br>• Hospital registers |
| Adoption | The number, proportion, representativeness of settings and intervention agents | • Provider and hospital characteristics<br>• System Use | • Provider surveys<br>• Facility assessment forms |
| Implementation | The intervention fidelity to the various elements of an intervention's protocol | • SMS delivery/receipt<br>• Provider engagement<br>• Changes to HITS2.1 | • SMS Log<br>• Surveys<br>• Field notes, IRB amendments |
| Maintenance | The extent to which a program or policy becomes part of routine practice | • Continued HITS 2.1 programmatic enrollment<br>• Cost effectiveness | • HITS 2.1 vs hospital registers<br>• Field notes |

rates, and patient volume to assess variations in cost effectiveness outcomes driven by key epidemiologic or demographic data.

The primary driver of the proposed cost-effectiveness analyses for HITSystem 2.1 will be the estimated reduction in the proportion of infants diagnosed with HIV at the initial test at 6-weeks of age. Since maternal nonadherence to ART, long turn-around time for VL results, and failure to take clinical action based on detectable VL results represent wasted investments in PMTCT services, secondary drivers of the cost-effectiveness analyses will be the proportions of women with optimal ART adherence during the antenatal and post-partum periods, women with VL results returned in time for prompt clinical action (by 36 weeks gestation), and detectable VL results that trigger guideline-adherent clinical action.

While the most rigorous plan is to use the number of perinatal transmissions observed in the study (intervention v control), we recognize the actual numbers are expected to be low. Thus, a secondary plan is to rely on viral suppression data in each study arm to estimate and compare perinatal transmission on a larger scale. Based on literature assessments of maternal viral load and perinatal HIV transmission of HIV [17, 29–32]. we will estimate rates of transmission at 2.5% and 22.5% for virally suppressed and unsuppressed women, respectively. We will use the estimated number of transmissions in intervention and control sites to compute savings from pediatric infections averted (lifetime ART costs for infants, HIV-related mortality). Data collected on ART adherence and VL testing will be used to calculate the wastage saved by HITSystem 2.1. VL test costing data have been assessed at the national level [33], and costing estimates for maternal and pediatric ART are available by regimen [34, 35], but outcome data from the proposed study are needed to adequately calculate the benefits of HITSystem 2.1. Anticipated costs, costs savings, and benefits are outlined in Table 4.

## Study oversight

Study-related adverse events will be reported to both overseeing IRBs, the study sponsor, and an established data and safety monitoring board (DSMB). Annual reviews by both IRBs will be conducted to ensure participant safety and compliance with study protocols. The external DSMB will be established to review and evaluate the accumulated study data for participant safety, study conduct, and progress on an annual basis. The DSMB will also conduct a blinded interim analysis after 67% of the planned sample in each arm has reached the endpoint for the primary outcomes to assess if HITSystem 2.1 efficacy can be determined for the main study outcomes: complete PMTCT retention (Aim 1) and maternal viral suppression (Aim 2) before accrual is complete. Members of the DSMB include an epidemiologist, a biostatistician, and a medical doctor with expertise in gynecology and obstetrics in global health settings. Any decisions for discontinuing or modifying allocated interventions will be discussed with the DSMB prior to implementation.

## Dissemination policy

Relevant new information on best practices for preventing perinatal HIV transmission that emerges from this study will be disseminated widely to all key stakeholders, including hospital,

Table 4. Anticipated costs, savings and benefits for cost-effectiveness analysis.

| HITSystem 2.1 Intervention Costs | Anticipated Cost Savings | Anticipated Benefits |
|---|---|---|
| Training, licensing, Internet, computer/modem, implementation support<br>*costs to health system may increase with higher rates of VL testing | Reduced: delayed/unreturned VL results, detectable VL results without clinical action, poor maternal ART adherence, pediatric infections and associated costs | QALY saved (infants)<br>QALY saved (mothers) |

county, and national level PMTCT and EID administrators and policy makers in Kenya; PMTCT and EID researchers and programmers, globally; and the community of caregivers of HIV-positive pregnant and postpartum women in Kenya. We will achieve these goals with the following steps: (1) the study registration will be updated annually during the course of the study to recognize progress and report results, (2) as data become available and are analyzed, results will be widely presented at national and international conferences and through publication of manuscripts, (3) study findings will be presented at dissemination meetings in Kenya to ensure that local, Kenyan hospital, county, and national level stakeholders (patients, clinicians, county administrators, and national policy makers) are informed of key results of the research, and (4) when appropriate, de-identified data will be deposited to public repositories. Key points of this dissemination plan are included on the informed consent documents, so that participants are informed of the plan prior to agreeing to participate in the research.

## Results

We are currently in year 2 of the 5-year study period and are actively enrolling participants at both intervention and control sites. Currently, only the first round of facility assessments have been conducted, and the first round of provider surveys are complete. To date, no data have been analyzed.

## Discussion

In spite of a global concerted effort, uptake and retention in quality, evidence-based PMTCT and HIV care remain suboptimal. This study will include vulnerable populations–HIV-infected mothers and HIV-exposed or HIV-infected infants living with severely constrained resources. Participation does have the potential for direct benefit. Complete PMTCT and guideline-adherent VL monitoring and clinical management have the potential to improve HIV+ women's health and prevent transmission of HIV to exposed infants. If transmission does occur, earlier linkage to infant HIV testing can allow for earlier treatment initiation, which has been shown to improve outcomes in HIV-infected infants. In addition, for those who miss their postpartum VL test, follow up efforts to obtain a sample and share VL results have the potential to directly benefit mother-infant pairs (especially if VL is high) by re-engaging them in care and facilitating action to address high viral load.

Findings from this study will make an important contribution to PMTCT services in Kenya and comparable low resource settings and will help address the outstanding need for evidence guiding implementation of eHealth strategies and viral suppression in this context. As Kenyan and global recommendations move toward eHealth and more stringent VL monitoring strategies, it is critical to assess the performance and cost-effectiveness of these methods in resource-constrained settings. These data will provide evidence for Ministries of Health and other organizations to consider optimal eHealth strategies to meet the global goal of eliminating new HIV infections in children through PMTCT services. We will work with participating counties and health facilities throughout the study period to develop sustainability plans beyond the course of the study. If efficacious, we will work with potential funders to facilitate the roll out of HITSystem 2.1 at health facilities throughout Kenya.

## Supporting information

**S1 Checklist. SPIRIT 2013 checklist: Recommended items to address in a clinical trial protocol and related documents**[*].
(PDF)

**S1 Appendix. Appendices 1–6 for baseline surveys, repeated surveys, facility assessment and provider survey.**
(PDF)

**S1 Protocol.**
(DOCX)

**S1 File. IRB approval (KEMRI).**
(PDF)

**S2 File. IRB approval (KUMC).**
(PDF)

## Acknowledgments

We would like to acknowledge members of the Kenya HITSystem team and partners: Kevin Oyowe, Anne Aloka, Caroline Otieno, Clarice Odhiambo, Dolleen Osundwa, Eddah Ongiri, Emma Ochieng, Everlyne Obwanda, Franscisca Obado, Irene Kasichana, Janice Kola, Loriet Morenya, and Priscilla Nyaiga. We also acknowledge the critical role of our government partners at NASCOP. We thank the director of KEMRI for their support.

## Author Contributions

**Conceptualization:** Sarah Finocchario-Kessler.

**Formal analysis:** Vincent Staggs.

**Funding acquisition:** Sarah Finocchario-Kessler.

**Methodology:** Vincent Staggs.

**Project administration:** Sharon Mokua, May Maloba, Catherine Wexler, Sarah Finocchario-Kessler.

**Writing – original draft:** Sharon Mokua, Catherine Wexler.

**Writing – review & editing:** Sharon Mokua, May Maloba, Catherine Wexler, Kathy Goggin, Vincent Staggs, Natabhona Mabachi, Nicodemus Maosa, Shadrack Babu, Emily Hurley, Sarah Finocchario-Kessler.

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
