## [Editor Report · Decision Letter 0]

7 Dec 2021

PONE-D-21-32121Evaluating the efficacy of the HITSystem 2.1 to improve PMTCT retention and maternal viral suppression in Kenya: Study protocol of a cluster-randomized trial.PLOS ONE

Dear Dr. Mokua,

Thank you for submitting your manuscript to PLOS ONE. After careful consideration, we feel that it has merit but does not fully meet PLOS ONE’s publication criteria as it currently stands. Therefore, we invite you to submit a revised version of the manuscript that addresses the points raised during the review process. During the internal evaluation of your manuscript we have noted the statement: "Since Kenyan law allows all pregnant females the same capacity to consent as an adult, young women under the age of 18 will be able to provide consent for themselves". Before we proceed further with your submission we will be grateful if you  could you also confirm whether the IRB have also agreed to waive parental consent for minors for this study

We look forward to receiving your revised manuscript.

Kind regards,

Lucinda Shen, MSc

Staff Editor

PLOS ONE

Journal Requirements:

3. Please ensure that you refer to Figure 2 in your text as, if accepted, production will need this reference to link the reader to the figure.
---

## [Author Response · Author response to Decision Letter 0]

21 Jan 2022

Dear editors and reviewers, 

Thank you for your review and feedback on our manuscript. Below, please find a item by item response to each of the queries raised:

1. During the internal evaluation of your manuscript, we have noted the statement: "Since Kenyan law allows all pregnant females the same capacity to consent as an adult, young women under the age of 18 will be able to provide consent for themselves". Before we proceed further with your submission we will be grateful if you could you also confirm whether the IRB have also agreed to waive parental consent for minors for this study.

Both the IRB at the University of Kansas Medical Center and the Kenya Medical Research Institute have reviewed the study protocol, including the capacity of pregnant women <18 years of age to consent for themselves, and approved it. 

We have reviewed the style requirements and believe our manuscript complies. 

The study was funded by: National Institute of Mental Health, Grant number: R01MH121245, awarded to Dr. Sarah Finocchario-Kessler. We have ensured this is correct in the manuscript and submission portal.

4. Please ensure that you refer to Figure 2 in your text as, if accepted, production will need this reference to link the reader to the figure.

We have made reference to the figures in the text.

Added

We have reviewed the reference list. While the reference list was complete, it had been synced with a previous version of the manuscript in EndNote; thus old reference 1 is now reference 24. We have also added the URLs for the web-based, non-journal references.

---

## [Editor Report · Decision Letter 1]

2 Feb 2022

Evaluating the efficacy of the HITSystem 2.1 to improve PMTCT retention and maternal viral suppression in Kenya: Study protocol of a cluster-randomized trial.

PONE-D-21-32121R1

Dear Dr. Mokua,

We’re pleased to inform you that your manuscript has been judged scientifically suitable for publication and will be formally accepted for publication once it meets all outstanding technical requirements.

Kind regards,

Lucinda Shen, MSc

Staff Editor

PLOS ONE
---

## [Editor Report · Acceptance letter]

8 Feb 2022

PONE-D-21-32121R1 

Evaluating the efficacy of the HITSystem 2.1 to improve PMTCT retention and maternal viral suppression in Kenya: study protocol of a cluster-randomized trial 

Dear Dr. Mokua:

I'm pleased to inform you that your manuscript has been deemed suitable for publication in PLOS ONE. Congratulations! Your manuscript is now with our production department. 

Kind regards, 

on behalf of

Miss Lucinda Shen 

Staff Editor

PLOS ONE